# Germanium-lead perovskite light-emitting diodes

Dexin Yang [1,2,6]✉, Guoling Zhang[1,6], Runchen Lai [1], Yao Cheng[2], Yaxiao Lian[1], Min Rao[2], Dexuan Huo[2], Dongchen Lan [3], Baodan Zhao[1,4,5] & Dawei Di [1,4,5]✉

Reducing environmental impact is a key challenge for perovskite optoelectronics, as most high-performance devices are based on potentially toxic lead-halide perovskites. For photo-voltaic solar cells, tin–lead (Sn–Pb) perovskite materials provide a promising solution for reducing toxicity. However, Sn–Pb perovskites typically exhibit low luminescence efficiencies, and are not ideal for light-emitting applications. Here we demonstrate highly luminescent germanium-lead (Ge–Pb) perovskite films with photoluminescence quantum efficiencies (PLQEs) of up to ~71%, showing a considerable relative improvement of ~34% over similarly prepared Ge-free, Pb-based perovskite films. In our initial demonstration of Ge–Pb perovskite LEDs, we achieve external quantum efficiencies (EQEs) of up to ~13.1% at high brightness (~1900 cd m$^{-2}$), a step forward for reduced-toxicity perovskite LEDs. Our findings offer a new solution for developing eco-friendly light-emitting technologies based on perovskite semiconductors.

[1] State Key Laboratory of Modern Optical Instrumentation, College of Optical Science and Engineering; International Research Center for Advanced Photonics, Zhejiang University, Hangzhou, China. [2] Key Laboratory of Novel Materials for Sensor of Zhejiang Province, College of Materials & Environmental Engineering, Hangzhou Dianzi University, Hangzhou, China. [3] College of Electrical Engineering, Zhejiang University, Hangzhou, China. [4] Key Laboratory of Excited-State Materials of Zhejiang Province, Zhejiang University, Hangzhou, China. [5] Cavendish Laboratory, University of Cambridge, Cambridge, United Kingdom. [6]These authors contributed equally: Dexin Yang, Guoling Zhang. ✉email: dy263@hdu.edu.cn; daweidi@zju.edu.cn

Metal-halide perovskites have emerged as a new class of semiconductor materials for next-generation display and lighting applications[1–6]. The combined advantages of tunable emission wavelengths, high spectral purity, high luminescence efficiencies, and low preparation costs are particularly attractive. Perovskite light-emitting diodes (PeLEDs) have exceeded the 20% external quantum efficiency (EQE) milestone in 2018[2–5], merely four years after the initial demonstration of room-temperature electroluminescence (EL) from halide perovskites[1]. Despite being excellent emitters showing near-unity internal quantum efficiencies (IQEs) for both EL[3] and photoluminescence (PL)[3,7], the toxicity of lead (Pb) hinders the development of perovskite light-emitting devices as an environmentally friendly emerging technology[6,8].

A useful and well-documented approach for reducing the toxicity of Pb in perovskite devices has been the use of tin (Sn) as a partial or complete replacement of Pb in the perovskite composition[9–17]. This strategy has been proven to be particularly successful in perovskite solar cells[12,14,18,19]. However, it has been widely observed that Sn-based (including Sn–Pb) perovskite materials show inferior light-emitting properties (Supplementary Table 1)[9,11,13,15,16,20–24] compared to Sn-free, Pb-based perovskites. This might be in part due to the reason that it is more likely to form a higher density of electronic defects and unsatisfactory film morphology related to the oxidation of $Sn^{2+}$ and rapid crystallization for Sn-based perovskites[9]. A recent report has shown decent EQEs of up to 5% from Sn-based perovskite LEDs[9], but the best EQE values occur at current densities ($<0.01$ mA cm$^{-2}$) below that is ideal for display applications (0.1–10 mA cm$^{-2}$).

In this work, we show that by using germanium (Ge), an environmentally friendly group-IV element, to partially substitute Pb in the perovskite precursor composition, it is possible to create highly luminescent perovskite materials and devices. We demonstrate Ge–Pb perovskite luminescent thin films with PL quantum efficiencies (PLQEs) of up to 71%, which is ~34% higher than that from a similarly prepared Ge-free, Pb-based perovskite composition (PLQE = 53%). Our initial demonstration of reduced-toxicity, Ge–Pb PeLEDs show best EQEs of ~13.1% at 4.68 mA cm$^{-2}$ (~1900 cd cm$^{-2}$), with a maximum brightness exceeding 10,000 cd m$^{-2}$. We show through PL experiments that the high luminescence efficiencies can be partly attributed to the enhanced radiative recombination in the Ge–Pb perovskites.

## Results

To create highly luminescent Ge–Pb perovskite films, we dissolved cesium bromide (CsBr), germanium bromide (GeBr$_2$), lead bromide (PbBr$_2$), phenylethylammonium bromide (PEABr), and molecular additive 1,4,7,10,13,16-hexaoxacyclooctadecane[25] in DMSO to form the precursor solution (see 'Methods' for details). A quasi-2D/3D mixed-dimensional perovskite composition[3,25,26] is expected to form after crystallization. The molar fraction of the Ge and Pb sources in the precursor solution, GeBr$_2$ and PbBr$_2$, are denoted as $x$ and $1-x$, where $0 < x < 1$. The perovskite precursor solution was spin-coated onto the substrates at 5000 rpm for 120 s. Ethyl acetate (200 μL) was drop-casted onto the spinning substrate 20 s after the start of the spin-coating process, followed by thermal annealing at 70 °C for 10 min.

The actual molar fractions of Ge in the resultant thin films are generally in line with that in the precursor solution, as confirmed by inductively coupled plasma optical emission spectroscopy (ICP-OES) (Fig. 1a). The molar fractions of Ge in the films are slightly higher than the intended values when the molar fractions of Ge precursor are equal to or less than 20% (For 10% and 20% Ge content in precursor, the corresponding Ge fractions in films are 15.0% and 25.7%, respectively). When the Ge fractions in precursor exceed 30%, the Ge fractions in films show very close agreement (Fig. 1a). Unless otherwise specified, for simplicity we refer to Ge fraction in precursors in the rest of the paper.

X-ray diffraction (XRD) analysis shows that the perovskite films with various levels of Ge incorporation adopt typical crystal structures of quasi-2D/3D perovskites (Fig. 1b)[3,26,27]. To obtain detailed crystal structure information of the perovskite films, Rietveld refinement[28] is used. The lattice parameters of the perovskite films as functions of the Ge molar fraction in the precursor solution are presented in Fig. 1c and Supplementary Fig. 1. As shown in Fig. 1c, the sample with 10% Ge inclusion shows the expected orthorhombic perovskite structure in the space group *Pbnm* (No. 62), with two octahedral tilt systems and a random distribution of Pb or Ge on the B-site. Figure 1d is the corresponding refined structure of the 3D orthorhombic crystal. The diffraction peaks at ~5° and ~10.5° are from the quasi-2D perovskite structure[29–31]. Based on the refinement results, the variation of the lattice parameters with different Ge molar fractions is shown in Fig. 1e and Supplementary Fig. 2. Significant variations of the lattice parameters $a$ and $b$ are observed when Ge ions are expected to partially replace Pb ions, exhibiting a typical octahedral distortion transition[32], while the fluctuations of the lattice parameters $c'$ of these samples are insignificant. The Ge–Pb perovskite films were analyzed using aberration-corrected scanning transmission electron microscopy (STEM). Perovskite nanocrystals in different regions of the film were inspected under the high-angle annular dark-field (HAADF) mode. The typical size of these nanocrystals is $8 \pm 1$ nm (Fig. 1f). The interplanar spacings of the crystalline structure observed in the high-resolution STEM-HAADF images are 3.5 Å and 4.1 Å (Fig. 1f), in agreement with the diffraction peaks at ~25° and ~21° of the XRD patterns, respectively (Fig. 1c).

The absorption and PL spectra of the perovskite films with 10% Ge inclusion are shown in Fig. 2a. The absorption edge and PL peak wavelengths are ~520 nm and ~515 nm, respectively, in agreement with what has been reported for quasi-2D/3D bromide perovskites[25,26]. While the XRD peak at ~5° (Fig. 1b) indicates the presence of quasi-2D perovskite PEA$_2$Cs$_{n-1}$(Pb$_{1-x}$Ge$_x$)$_n$Br$_{3n+1}$ ($n$ is the number of inorganic octahedral sheets between the organic spacers[6]), the typical absorption peaks (405, 436, and 467 nm) for the quasi-2D phases cannot be clearly observed for the samples with 10% Ge inclusion (Fig. 2b). The suppression of the absorption peaks from the quasi-2D perovskite might be due to the strong absorption of the dominant 3D phase that masks the absorption of quasi-2D phases. However, the absorption features of the quasi-2D phases become visible when the Ge inclusion reaches 40% (Fig. 2b). As shown in Fig. 2b and Supplementary Table 2, the optical bandgaps for the Ge–Pb perovskite films remain nearly constant[33,34], at around 2.4 eV, as estimated from the absorption spectra (Supplementary Table 2). Importantly, a Ge molar fraction of 10–20% enhances the PLQEs of the perovskite films from ~53% (Ge-free) to ~71% (Fig. 2c), in clear contrast to the case of Sn–Pb perovskites where even a small inclusion of Sn significantly reduces the PLQE (Supplementary Fig. 3). Further increase in Ge molar fraction to >30% reduces the PLQEs to below 40%.

The PLQE variation of the Ge–Pb perovskite samples is unlikely a result of optical scattering. As shown in Fig. 2d, the reference Ge-free sample and the sample with 10–20 mol% Ge both show smooth surface morphology as confirmed by scanning electron microscopy (SEM) (Fig. 2d and Supplementary Fig. 4a–c) and atomic force microscopy (AFM) (Fig. 2d and Supplementary Fig. 5a–c), showing average surface roughness of ~1.37 nm inside a typical field of view of 1 μm. The roughness of the perovskite films increases when the Ge molar fraction reaches

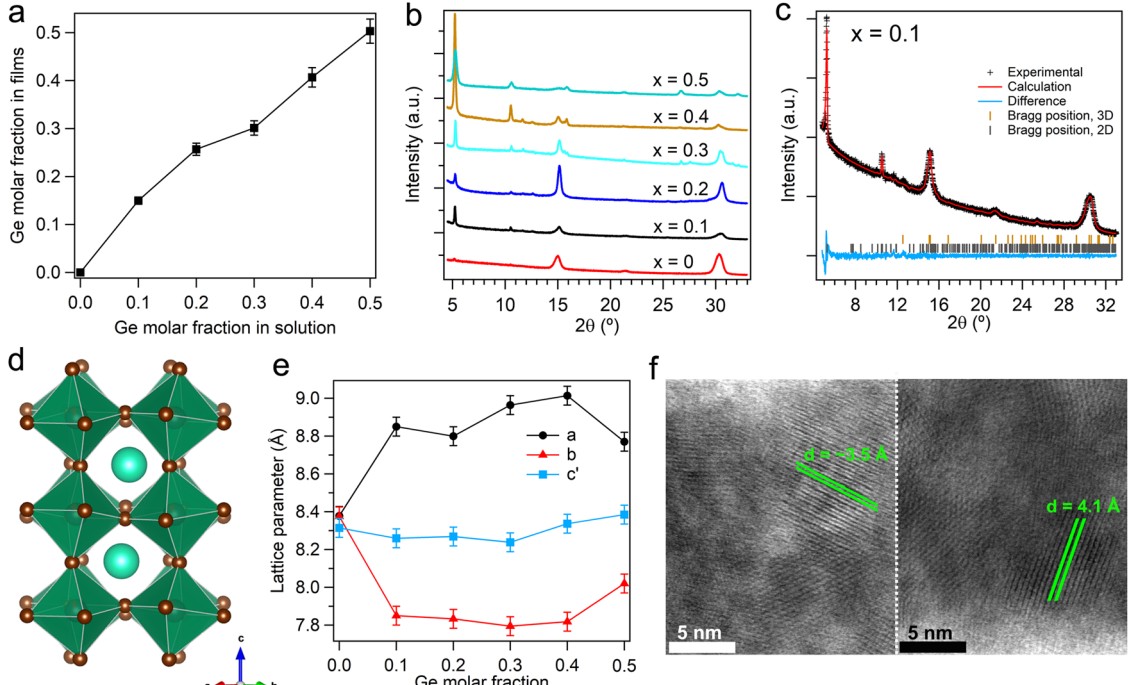

**Fig. 1 Structural and compositional characterization of the Ge–Pb perovskite samples. a** The ICP-OES results of Ge molar fraction in thin films versus the intended Ge content in the precursor solution. The error bars represent the estimated experimental errors. **b** The XRD results of the perovskite films deposited on Si substrates with different Ge molar fractions. **c** XRD pattern and the corresponding fitting using Rietveld refinement for the perovskite sample with 10 mol% Ge inclusion. The observed profile is marked by black crosses and the calculated profile is represented by the red line. Bragg peak positions of the 3D perovskite $CsPb_{0.9}Ge_{0.1}Br_3$ and 2D perovskite $PEA_2Pb_{0.9}Ge_{0.1}Br_4$ are labeled by orange and gray marks, respectively. The difference diffractogram (experimental minus calculated) is shown in light blue. **d** Refined crystal structure: green octahedra represent $Pb(Ge)Br_6$; Pb or Ge atoms are in the centers of the octahedra; blue-green spheres represent Cs atoms; brown spheres represent Br atoms. **e** Pseudo-cubic ($c' = c/\sqrt{2}$) lattice parameters variation for 3D perovskite $CsPb_{1-x}Ge_xBr_3$ ($x = 0, 0.1, 0.2, 0.3, 0.4$, and $0.5$). The error bars represent the estimated errors. **f** The STEM-HAADF images showing the structure of the perovskite nanocrystals with 10 mol% Ge inclusion from different regions.

30–50% (Supplementary Fig. 5d–f), but as we show below, the potential benefit of enhanced light out-coupling by the scattering surfaces does not outweigh additional nonradiative losses in these films.

Transient PL decay measurements were carried out to study the carrier recombination kinetics in the Ge–Pb perovskite samples. Figure 2e shows the PL decay curves under the minimum excitation intensity (10 nJ cm$^{-2}$). The effective PL lifetimes (the time required for the PL intensity to reach 1/e of the initial intensity), and the PL decay tail lifetimes[3] (the lifetimes of the tails of the decay traces where the excitation densities are assumed to reach minimum) of the samples are summarized in Fig. 2f. The PL decay tail lifetime decreases for Ge molar fractions of 10–20%, where the corresponding PLQE approaches its peak value (~71%). This is an unexpected result, as the tail lifetimes are commonly associated with the rates of trap-assisted non-radiative recombination and are normally longer for emitters with improved defect passivation[3,26]. A possible explanation would be that the ratio of radiative to non-radiative rates is higher for these Ge molar fractions, in agreement with the improved PLQEs. At the molar fraction of 30%, the PL decay tail lifetime reaches the minimum value of ~200 ns, which is accompanied by a steep fall of PLQE to 35%. The tail lifetime and effective lifetime generally increase at a Ge molar fraction of 40%, where the PLQE continues to decrease to ~32%. Overall, our observations are consistent with the view that at an optimal Ge molar fraction of 10–20% for these samples, radiative recombination of excited states dominates over non-emissive processes, leading to the high PLQEs. Beyond a Ge molar fraction of >30%, trap-assisted recombination starts to dominate, resulting in reduced PLQEs.

To gain further insights into the emission mechanisms, we conducted excitation-power-dependent PL measurements for the Ge–Pb perovskite films. The representative PL decay curves are presented in Fig. 2g. The normalized PLQEs as functions of excitation fluence is shown in Fig. 2h. As summarized in Supplementary Figs. 6 and 7, the effective lifetimes of the perovskite films reduce as the excitation power density increases from 10 nJ cm$^{-2}$ to 20 µJ cm$^{-2}$. The power dependence of effective lifetimes (Supplementary Figs. 6 and 7) is consistent with the PLQE variations shown in Fig. 2h. For the samples with Ge molar fractions of 0–20%, the generally flat PL lifetime and PLQE curves at excitation fluences below 100 nJ cm$^{-2}$ indicate an excitonic character of the emissive species. The PLQEs reduce at higher fluences, in line with possible Auger-like processes[35–37] under these excitation conditions. More detailed spectroscopic studies are required to establish a comprehensive view of the recombination dynamics, but they are beyond the scope of the present work. Limited by the maximum orders of magnitude variation of our transient PL decay measurements, the apparent tail lifetimes of all the perovskite films generally decrease as the excitation fluence increases (Supplementary Fig. 6). Therefore the tail lifetimes measured here may not offer an accurate reflection of the density of non-radiative traps. To address this experimental limitation, we carried out further electrical measurements to understand the role of charge traps (*vide infra*). Besides, we find that Ge inclusion generally improves the stability of the perovskite films under optical excitation, as shown in Fig. 2i and Supplementary Fig. 8.

The smooth and highly luminescent Ge–Pb perovskite films may be well suited for LED applications. As an initial demonstration, we

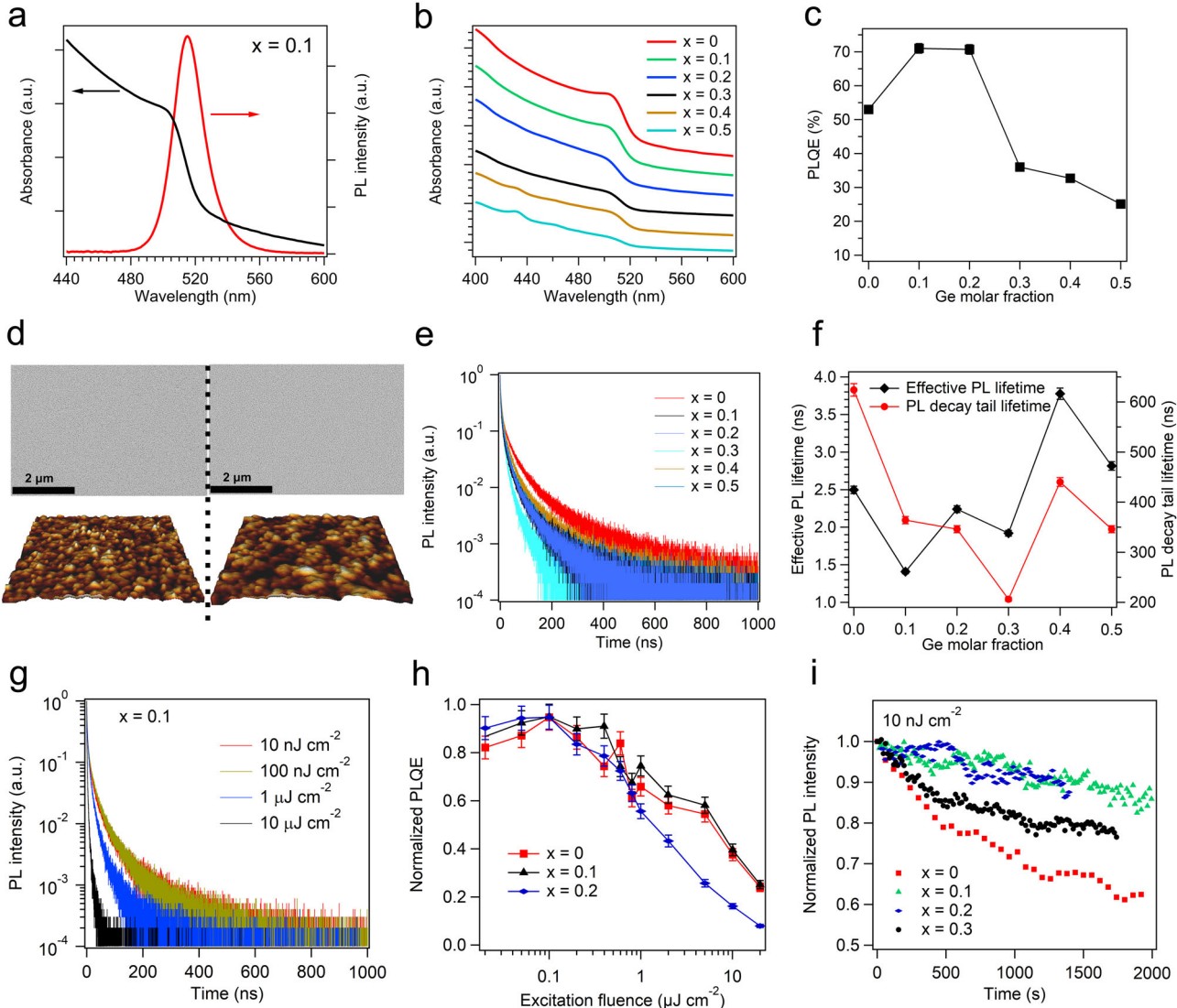

**Fig. 2 Optical properties and surface morphology of the Ge–Pb perovskite samples. a** Absorption and PL spectra of the perovskite film with 10 mol% Ge inclusion. **b** Absorption spectra of the Ge–Pb perovskite films with Ge molar fractions of 0–50%. **c** PLQE versus Ge molar fraction for the perovskite films. **d** SEM and AFM images of the perovskite films without and with 10 mol% Ge inclusion. **e** Transient PL decay profiles of the Ge–Pb perovskite films under 10 nJ cm$^{-2}$ excitation. **f** Effective PL lifetimes and PL decay tail lifetimes of the Ge–Pb perovskite samples. The error bars represent the estimated experimental errors. **g** Transient PL decay profiles for the sample with 10% Ge inclusion under excitation intensities ranging from 10 nJ cm$^{-2}$ to 20 μJ cm$^{-2}$. **h** Normalized PLQEs for the Ge–Pb perovskite films with Ge molar fractions of 0–20%. The error bars represent the estimated experimental errors. **i** PL stability measurements for Ge–Pb perovskite films under pulsed excitation (10 nJ cm$^{-2}$).

prepared Ge–Pb PeLED devices using a simple device structure of ITO/poly(3,4-ethylene dioxythiophene): polystyrene sulfonic acid (PEDOT:PSS)/poly(9-vinylcarbazole)(PVK)/perovskite/2,2′,2″-(1,3,5-benzinetriyl)-tris(1-phenyl-1-*H*-benzimidazole) (TPBi)/LiF/Al, as shown in Fig. 3a. Figure 3b shows the energy level diagram of the Ge–Pb PeLEDs. The energy levels of the perovskite films are characterized using ultraviolet photoemission spectroscopy (UPS) (Supplementary Figs. 10 and 11) and are summarized in Supplementary Table 2. A Ge–Pb PeLED with 10 mol% Ge was analyzed using STEM under the HAADF mode (Fig. 3c). Figure 3d shows the energy dispersive spectroscopy (EDS) elemental maps of Cs, Pb, and Ge in the same area enclosed by the dashed box in Fig. 3c, confirming a uniform distribution of the elements in the Ge–Pb perovskite emissive layer.

Figure 3e shows the EL spectra of a Ge–Pb PeLED with 10 mol% Ge inclusion under a range of driving voltages. The devices exhibit spectrally stable EL centered at around 514 nm with an FWHM of

~20 nm. Inset shows a photograph of the working PeLED. The maximum EQE of the Ge–Pb PeLEDs with 10 mol% Ge is 13.1% at 4.68 mA cm$^{-2}$ (~1900 cd m$^{-2}$), a considerable improvement over the Ge-free PeLED control devices with a peak EQE of ~11.3% (Fig. 3f). The efficiency improvement may be attributed to the reduced trap density (estimated hole trap density: $1.06 \times 10^{18}$ cm$^{-3}$, electron trap density: $5.72 \times 10^{17}$ cm$^{-3}$) of the Ge–Pb PeLEDs with 10 mol% Ge compared to the control devices (hole trap density: $1.11 \times 10^{18}$ cm$^{-3}$, electron trap density: $8.06 \times 10^{17}$ cm$^{-3}$), as suggested by space-charge-limited current (SCLC) analysis[38,39] (Supplementary Figs. 11a, b and 12a, b). However, as the Ge substitution further increases (20–50 mol% Ge), the trap densities of those Ge–Pb perovskite films increase (e.g., for a Ge inclusion of 30%, the hole trap and electron trap densities are $1.53 \times 10^{18}$ cm$^{-3}$ and $1.22 \times 10^{18}$ cm$^{-3}$, respectively), consistent with the markedly reduced PLQEs of these samples (Supplementary Figs. 11c–f and 12c–f).

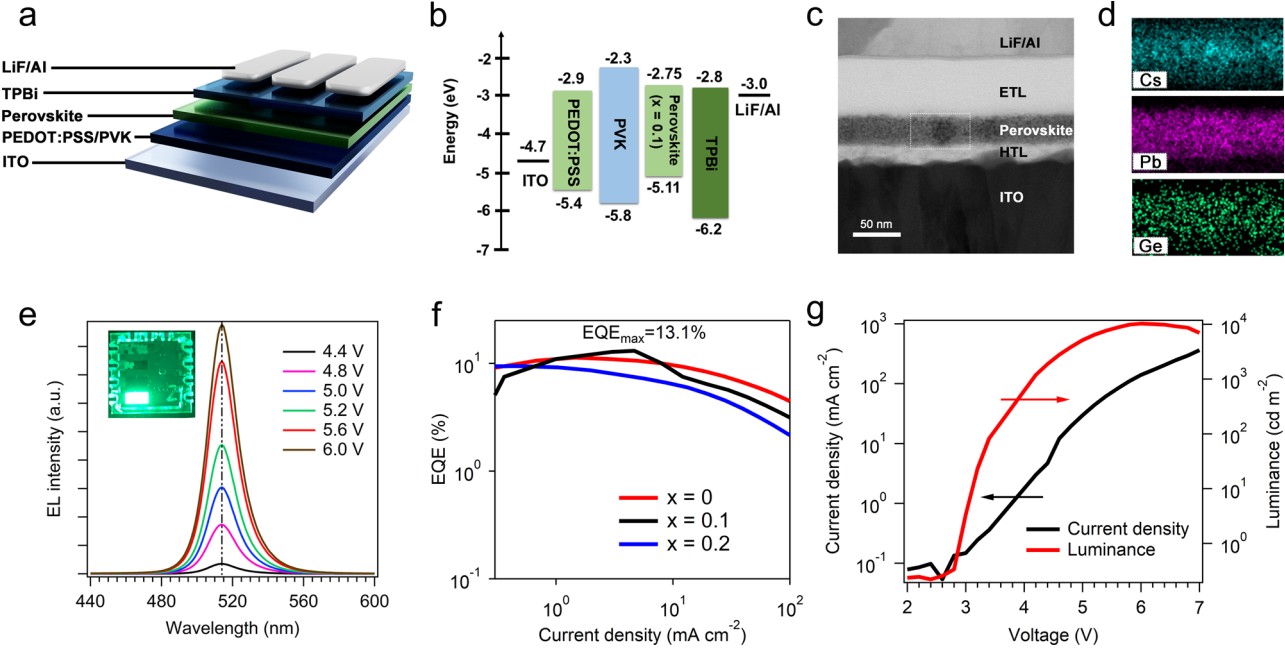

**Fig. 3 Characterization of Ge–Pb PeLED performance. a** Schematic of the Ge–Pb PeLED device structure. **b** Flat-band energy level diagram of a Ge–Pb PeLED. **c** A STEM-HAADF image of the cross-section of a Ge–Pb PeLED. **d** EDS elemental maps of Cs, Pb, and Ge in the same area enclosed by the dashed box in (**c**). **e** The EL spectra across a range of operating voltages. Inset: a photograph of a working Ge–Pb PeLED. **f** EQE-current density curves for PeLEDs with 0 mol%, 10 mol%, and 20 mol% Ge inclusion. **g** Current density-luminance-voltage characteristics of the best-performing PeLED with 10 mol% Ge inclusion.

In contrast to the improved PL stability for films under optical excitation (Fig. 2i and Supplementary Fig. 8), the stability of our Ge–Pb PeLEDs with 10 and 20 mol% Ge content ($T_{50}$ for EL: 18 and 10 min at 1 mA cm$^{-2}$) is inferior to that of the Ge-free control devices ($T_{50}$ for EL: ~30 min) (Supplementary Fig. 13a). Supplementary Fig. 13b shows the peak EQE histogram of Ge–Pb PeLEDs with 10 mol% Ge inclusion. The maximum luminance obtained from the Ge–Pb PeLEDs is ~10,000 cd m$^{-2}$ (Fig. 3g). The peak EQE and luminance achieved from these devices represent a record for reduced-toxicity PeLEDs (Supplementary Table 2).

## Discussion

In summary, we have developed highly luminescent Ge–Pb perovskite films with PL efficiencies of up to 71%, showing a considerable relative improvement of ~34% over similarly prepared Ge-free, Pb-based perovskite films. In our initial demonstration of Ge–Pb PeLEDs, we achieve EQEs of up to 13.1% at high brightness (~1900 cd m$^{-2}$), a record for reduced-toxicity PeLEDs. In contrast to Sn-based perovskite materials which suffer from dominant non-radiative recombination losses, Ge inclusion at a suitable molar fraction enhances luminescence efficiencies. While further optimization is required for improving the device performance, our results open a promising route toward eco-friendly light-emitting technologies based on perovskite semiconductors.

## Methods

**Preparation of perovskite precursor solution and films**. The perovskite precursor solution was prepared by dissolving lead bromide (PbBr$_2$) (99.999%, Sigma-Aldrich), cesium bromide (CsBr) (99.9%, Alfa Aesar), germanium dibromide (GeBr$_2$) (97%, Sigma-Aldrich), 24 mg 2-phenylethylammonium bromide (PEABr) (>99.5%, Xi'an polymer Light), and 5 mg 1,4,7,10,13,16-hexaoxacyclooctadecane (99%, Sigma-Aldrich) in 1 mL dimethylsulfoxide (DMSO) (Sigma-Aldrich) in an argon-filled glovebox. The solution was stirred overnight at room temperature. Perovskite films were obtained by spin-coating precursor solution onto the precleaned fused silica substrates at 5000 rpm for 120 s in an argon-filled glovebox. Ethyl acetate (200 μL) was drop-casted onto the spinning substrate 20 s after

the start of the spin-coating process. The resultant film was annealed at 70 °C for 10 min.

**Fabrication of PeLEDs**. Pre-patterned indium tin oxide (ITO)-coated glass substrates (15 ohms/square) were cleaned using ultra-sonication in acetone and isopropanol for 15 min, respectively. The substrates were dried with a nitrogen blowgun, followed by ultraviolet ozone treatment (LEBO science, UC100). PEDOT: PSS (Clevios P VP AI 4083) was spin-coated onto the ITO-coated glass substrates at 4000 rpm and was annealed at 150 °C for 20 min. The ITO/PEDOT:PSS substrates were then transferred to an argon-filled glovebox. PVK was spin-coated from chlorobenzene (CB) solution (8 mg mL$^{-1}$) at 5000 rpm and was annealed at 120 °C for 30 min. Subsequently, the perovskite film (~30 nm) was spin-coated from the precursor solution at 5000 rpm for 120 s. Ethyl acetate (200 μl) was drop-casted onto the spinning substrate 20 s after the start of the spin-coating process. The resultant films were annealed at 70 °C for 10 min. Finally, TPBi (50 nm), LiF (1 nm), and Al (80 nm) were sequentially evaporated through a shadow mask at a pressure of 10$^{-6}$ mBar. All the devices were encapsulated with UV epoxy (NOA81, Thorlabs)/cover glass to minimize exposure to oxygen and moisture during measurements.

**AFM measurements**. Topographic images of the perovskite films on fused silica substrates were obtained by atomic force microscopy (JPK nanoWizard 4-NanoScience).

**SEM measurements**. SEM measurements were carried out on perovskite films deposited on silicon substrates using a high-resolution scanning microscope (Apreo S, FEI).

**High angle annular dark-field scanning transmission electron microscopy (HAADF-STEM)**. The EDS mapping and cross-sectional information of the PeLEDs were investigated by HAADF-STEM using an FEI Titan G2 80–200 ChemiSTEM microscope equipped with an aberration corrector for probe forming lens, operated at 200 kV. The LED cross-section samples were prepared by using a dual-beam focused-ion-beam system (Quata 3D FEG).

**ICP-OES measurements**. To obtain accurate stoichiometric ratios between the Ge and Pb elements in the perovskite films, the concentrations of the two species were determined using the ICP-OES (Agilent 720ES). Sample preparation details: the perovskite films on fused silica substrates were obtained as described in the film preparation section. The samples were dipped into water with ultrasonic treatment until the perovskite films are completely dissolved. The resultant solution was used for the ICP-OES measurements.

**XRD measurements.** XRD measurements were carried out on perovskite films on silicon substrates using a Rigaku-SmartLab (9 kW) X-ray diffractometer with Cu $K\alpha_{1,2}$ radiation ($\lambda = 1.541$ Å). Spectra were collected with an angular range of $4° < 2\theta < 35°$. TOPAS-Academic V6 software[28] was used for Rietveld refinements to obtain the lattice parameters. The background and peak shapes were fit using a shifted Chebyshev function with eight parameters and a Pseudo-Voigt function (TCHZ type), respectively.

**PLQE and PL measurements.** The PLQE and PL of thin-film samples were measured using an integrating sphere-spectrometer setup[40], in air. A continuous-wave 405-nm diode laser (excitation intensity: ~9 mW cm$^{-2}$) was used as the source of excitation. The PL spectra were collected using an Ocean Optics spectrometer (USB4000).

**UPS measurements.** UPS was used to determine the work function and valence band maximum of perovskite films. The UPS spectra were acquired in an ultrahigh-vacuum chamber with excitation provided by the He I emission line (21.2 eV) of a helium discharge lamp. The VBM and binding energies are assigned on the basis of a Gaussian fit to the UPS onset region[41,42], and the WF is calculated by $\Phi = 21.2 - $onset[26], as shown in Supplementary Figs. 3 and 4 and Supplementary Table 2 of all perovskite samples. The UPS experiments were performed using Thermo Scientific ESCALAB 250Xi.

**Time-correlated single-photon counting (TCSPC) and PL stability measurements.** The perovskite thin films (on fused silica substrates) were excited by a 400-nm femtosecond laser (pulse duration ~150 fs) generated from an optical parametric amplifier (OPA, Orpheus-F, Light Conversion) pumped by a Yb$^+$:YAG femtosecond laser (~270 fs, 50 kHz; Pharos, Light Conversion). The excitation beam was focused by a lens onto the perovskite films from the substrate side, and the PL was collected by an objective (SOPTOP, LMPlan 10*, NA = 0.3) from the opposite side. The PL after a long-pass filter (FELH450, Thorlabs) was sent through a 50/50 beam splitter and the transmitted light was delivered into a fiber-coupled single-photon avalanche photo-diode (APD; ID100, IDQ) and the time-resolved decay curves were collected by a PicoHarp 300 counter (PicoQuant). PL photons into the APD were attenuated so that the PL count rates were less than 5% of the excitation frequency. The time resolution of the TCSPC system is ~200 ps (from the FWHM of the instrumental response function). The reflected light was focused onto a multi-mode fiber and delivered to an Ocean HDX spectrometer (Ocean Optics) for PL stability measurements. The excitation light spot was ~680 μm in diameter determined by the knife-cutting method. The detection region for TCSPC measurements was restricted to the central area of the excitation spot. The samples were sealed in an argon-filled chamber with fused silica windows once after fabrication and during the measurements.

**Characterization of LED performance.** Current density–voltage ($J$–$V$) characteristics were measured using a Keithley 2400 source-meter unit. The luminance and EQE data were obtained using an Everfine OLED-200 commercial OLED performance analysis system. The photon flux and EL spectra were measured using a charge-coupled device centered over the light-emitting pixel. The luminance (in cd m$^{-2}$) of the devices was calculated based on the emission function of the PeLED and on the known spectral response of the charge-coupled device, and the EQEs of the devices were calculated assuming a Lambertian emission profile. In order to determine the trap density of Ge–Pb perovskite films prepared with different Ge substitutions, we carried out space charge limited current (SCLC) analysis using the equation[38,39]: $N_t = \frac{2\varepsilon_0 \varepsilon V_{TEF}}{eL^2}$, where $\varepsilon_0$ is the vacuum permittivity, $\varepsilon$ is the relative permittivity of perovskite, $V_{TEF}$ is the trap-filled-limit voltage, $e$ is the elementary charge, and $L$ is the thickness of the thin film. The film thickness (from HAADF-STEM studies) and relative permittivity of all Ge–Pb perovskite films were estimated to be 50 nm and 29.37[43], respectively. More accurate determination of these materials parameters may lead to further refined results.

## Data availability

The research data supporting this paper are available from the corresponding authors upon reasonable request.

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

## Acknowledgements

This work was supported by the National Key R&D Program of China (grant no. 2018YFB2200401), the National Natural Science Foundation of China (NSFC) (61975180, 51702289, 62005243), Kun-Peng Program of Zhejiang Province (D.D.), Natural Science Foundation of Zhejiang Province (LR21F050003), China Postdoctoral Science Foundation (2020M681817), Fundamental Research Funds for the Central Universities (2019QNA5005, 2020QNA5002), and Zhejiang University Education Foundation Global Partnership Fund. We acknowledge the technical support from the Core Facilities, State Key Laboratory of Modern Optical Instrumentation, Zhejiang University.

## Author contributions

D.Y. planned the project under the guidance of D.D. G.Z. and D.Y. developed and characterized the efficient Ge–Pb PeLEDs. Y.L. and Y.C. contributed to the device fabrication. D.Y., G.Z. and Y.C. developed the luminescent Ge–Pb perovskite thin films and carried out the steady-state PL and absorption experiments. D.L. contributed to the analysis of the optical absorption data. R.L. and D.Y. conducted the power-dependent transient PL and PL stability experiments, with data analysis carried out by R.L. D.Y. performed XRD, SEM, ICP-OES, and AFM experiments. Y.C. and M.R. contributed to sample preparation and measurements under the guidance of D.H. and D.Y. D.Y. carried out crystalline structure analysis using Rietveld refinement and performed the HAADF-STEM and UPS studies. D.Y. wrote the initial manuscript, which was revised by D.D. and B.Z. All authors contributed to the work and commented on the paper.

## Competing interests

D.D. and D.Y. are inventors on CN patent application: 202110103577.5. The remaining authors declare no competing interests.
