## [Peer Review File · Nature Communications]

REVIEWER COMMENTS

Reviewer #1 (Remarks to the Author):

Yang and collaborators report on the performance of light-emitting diodes based on hybrid perovskite nanocrystals where Pb atoms have been partially replaced by Ge. The reported LED efficiencies are noteworthy, as is the attempt to reduce the amount of poisonous lead in the devices.

I feel however there is a major shortcoming in the manuscript: the authors do not establish the composition of their nanocrystals, citing only the molar fraction of Ge and Pb precursors. Nothing however guarantees that the nanocrystal composition reflects the molar ratio of precursors, nor that it is simply proportional to it, as the amount of Ge actually entering the nanocrystal composition may be determined by the kinetics of the different reactions involved. Since replacing lead with Ge is really the main claim of the manuscript, an additional effort is needed to measure the actual Ge fraction in the nanocrystals.

Another criticism, more specific this time, is that the measured photoluminescence quantum yields seem at odds with the measured photoluminescence lifetimes under pulsed excitation. If the quantum yield approaches unity, the decay is mainly due to radiative recombination, is mostly exponential and a larger quantum yield should correspond to a proportionally faster decay. The authors should try to explain why this does not seem to happen and rule out any possible systematic error in measurements, which is easy to incur in when the measurements output a single number, like in absolute quantum yield measurements.

I recommend that the issues are addressed before considering publication.

Reviewer #2 (Remarks to the Author):

The work "Germanium-lead perovskite light-emitting diodes" described the development of the germanium-lead perovskite films and the light-emitting diodes based on them. The importance of this work is not clear because the toxic Pb was still used, and the lifetime of the LEDs is still unknown. The novelty is limited and this work might not be suitable for Nature Communications.

1. AFM images of other x values are need. Comparison between these images are required.
2. Please identify the layers shown in Figure 3d just like Figure 3c.
3. Please provide the statistics of the performance of PeLEDs.
4. Please provide the lifetime of the device under constant current density or constant voltage.
5. In-depth analysis based on optical and electrical measurements are required to explain the best doping ratio.

Reviewer #3 (Remarks to the Author):

In this article, Dawei Di and coworkers used Ge-Pb based green perovskites to produce eco-friendly perovskite LEDs. While it is important to reduce the Pb-content to address the toxicity, I believe the data shown here do not provide a concrete evidence on reduced Pb in their perovskite structures. In addition, the EQEs are substantially lower compared to many other literature articles. Therefore, I would recommend a major revision before the publication in Nature Communications.

My comments are given below

1. There is no experimental evidence on the presence of 30% Ge in their perovskites. From XRD we overestimate this data. Therefore, I would recommend authors to perform rigorous elemental analyses (ICP-MS/OES or AAS) before claiming 30% Ge.
2. I am highly doubtful that with presence of 30% Ge, the perovskite can still hold its crystal structure. I would derive the tolerance factor to make sure the perovskite is still holding its orthorhombic structure and supporting XRD.
3. From Figure 2e), why slower decay in 40% Ge samples, although the PLQY is low, compared to samples with 20% is unclear. I would recommend the authors to present this at various incident power densities and see if the trends change. I appreciate that the authors admit this discrepancy,

however, providing more insights on decay kinetics will certainly add a value to this paper.

4. Why EQEs of their control devices are lower compared to equivalent LEDs in the literature?

5. Stability is an important factor. So, I would recommend authors to present operational stability with time, and curious to see how it changes when Ge is present.

Response to Review Comments

Reviewer #1 (Remarks to the Author):

Yang and collaborators report on the performance of light-emitting diodes based on hybrid perovskite nanocrystals where Pb atoms have been partially replaced by Ge. The reported LED efficiencies are noteworthy, as is the attempt to reduce the amount of poisonous lead in the devices.

R: We thank the reviewer for the supportive comments.

I feel however there is a major shortcoming in the manuscript: the authors do not establish the composition of their nanocrystals, citing only the molar fraction of Ge and Pb precursors. Nothing however guarantees that the nanocrystal composition reflects the molar ratio of precursors, nor that it is simply proportional to it, as the amount of Ge actually entering the nanocrystal composition may be determined by the kinetics of the different reactions involved. Since replacing lead with Ge is really the main claim of the manuscript, an additional effort is needed to measure the actual Ge fraction in the nanocrystals.

R: We thank the reviewer for raising this important point. We agree that it is not possible to establish the composition of the films by simply citing the molar fractions of Ge and Pb in the precursor solution. To address this point, we performed inductively coupled plasma optical emission spectroscopy (ICP-OES) measurement to determine the actual Ge fraction in the solid films. The results are shown in Fig. 1a in the revised manuscript. In the revised text (page 2-3), we clarify that:

The actual molar fractions of Ge in the resultant thin films are generally in line with that in the precursor solution, as confirmed by inductively coupled plasma optical emission spectroscopy (ICP-OES) (Figure 1a). The molar fractions of Ge in the films are slightly higher than the intended values when the molar fractions of Ge precursor are equal to or less than 20% (For 10% and 20% Ge content in precursor, the corresponding Ge fractions in films are 15.0% and 25.7%, respectively). When the Ge fractions in precursor exceed 30%, the Ge fractions in films show very close agreement.

Another criticism, more specific this time, is that the measured photoluminescence quantum yields seem at odds with the measured photoluminescence lifetimes under pulsed excitation. If the quantum yield approaches unity, the decay is mainly due to radiative recombination, is mostly exponential and a larger quantum yield should correspond to a proportionally faster decay. The authors should try to explain why this does not seem to happen and rule out any possible systematic error in measurements, which is easy to incur in when the measurements output a single number, like in absolute quantum yield measurements.

I recommend that the issues are addressed before considering publication.

R: We thank the reviewer for the helpful comment. In the revised manuscript, we have carefully carried out additional optical measurements (see updated Figure 2, and Supplementary Fig. 6-8) to ensure the reliability of our results. We found that the PL decay lifetime generally reduces for Ge molar fractions of 10-20%, where the corresponding PLQE approaches its peak value (~71%). Our observations are consistent with the view that at an optimal Ge molar fraction of 10-20% for these samples, radiative recombination of excited states dominates over the non-emissive processes, leading to the maximum PLQEs. Beyond a Ge molar fraction of >30%, trap-assisted recombination starts to dominate, resulting in reduced PLQEs.

In addition, we have carried out further excitation-intensity-dependent optical measurements to understand the emission mechanisms. We found that for the samples with Ge molar fractions of 0-30%, the generally flat PL lifetime and PLQE curves at excitation fluences below 100 nJ cm⁻² indicate an excitonic character of the emissive species. The PLQEs reduce at higher fluences, in line with possible Auger-like processes under these conditions. Relevant discussions are included in the revised manuscript (page 5-6, highlighted in blue).

Reviewer #2 (Remarks to the Author):

The work “Germanium-lead perovskite light-emitting diodes” described the development of the germanium-lead perovskite films and the light-emitting diodes based on them. The importance of this work is not clear because the toxic Pb was still used, and the lifetime of the LEDs is still unknown. The novelty is limited and this work might not be suitable for Nature Communications.

R: We thank the reviewer for the criticism and constructive comments, which have encouraged us to carry out substantial revisions to improve the paper.

So far, the best-performing perovskite LEDs are from lead-halide precursors. Discovering reduced-toxicity perovskite light sources is a key challenge in this emerging field. For perovskite solar cells, tin-lead (Sn-Pb) perovskites provide a promising solution for reducing toxicity. However, Sn-Pb perovskites typically exhibit very low luminescence efficiencies, and are not ideal for light-emitting applications (see Supplementary Table 1 and Supplementary Figure 3). In this work, we demonstrate highly luminescent germanium-lead perovskite films and devices. In contrast to Sn-based perovskite materials which suffer from dominant non-radiative losses, Ge inclusion at a suitable concentration enhances luminescence efficiencies. We have clarified the importance of our work in the abstract, introduction and conclusion sections of the revised manuscript.

Following the reviewer’s suggestion, we have measured the operational lifetimes of these LEDs, and the results are presented in Supplementary Figure 13a in the revised paper. We note that the device half-life (T_{50}) of 10-18 min at 1 mA/cm² is far from satisfactory for practical applications, but we hope the reviewer would agree that the substantially revised paper represents a useful step toward eco-friendly perovskite

LEDs.

1. AFM images of other x values are needed. Comparison between these images are required.

R: Following the reviewer's suggestion, the AFM images of other x values have been included in the revised manuscript as Supplementary Figure 5. A brief discussion on the results is included on page 5 of the revised manuscript.

2. Please identify the layers shown in Figure 3d just like Figure 3c.

R: We thank the reviewer for the suggestion. Figure 3d shows the EDS elemental distributions of Cs, Pb, and Ge in the same area (dashed box) in Figure 3c. We have now clarified this issue in the caption of Figure 3d. The EDS results confirm that Cs, Pb and Ge elements distribute homogeneously in the perovskite thin films.

3. Please provide the statistics of the performance of PeLEDs.

R: Following the reviewer's comment, we have now included the statistics of the EQEs of the PeLEDs (Supplementary Figure 13b) in the revised supplementary information file. In addition, we have updated the champion PeLED performance data as a result of improved device optimization during the revisions.

4. Please provide the lifetime of the device under constant current density or constant voltage.

R: We thank the reviewer for the suggestion. The device operational lifetimes under a constant current density is shown in Supplementary Figure 13a in the revised supplementary information file. Under a current density of 1 mA/cm², encapsulated devices with 10%-20% Ge inclusion show half-life (T_{50}) of 10 -18 min in air. Improving the device stability is an important subject of our future research.

5. In-depth analysis based on optical and electrical measurements are required to explain the best doping ratio.

R: We appreciate the helpful advice from the reviewer. Following the suggestion, to explain the best doping ratio we have carried out many sets of additional optical and electrical experiments during the revision process. See Supplementary Fig. 6-8, and updated Figure 2 for additional optical measurements; Supplementary Figures 11 and 12 for additional electrical measurements. Our observations are consistent with the view that at an optimal Ge molar fraction of 10-20% for these samples, radiative recombination of excited states dominates over the non-emissive processes, leading to the maximum PLQEs. Beyond a Ge molar fraction of >30%, trap-assisted recombination starts to dominate, resulting in reduced PLQEs.

For example, the transient PL decay measurements of the Ge-Pb perovskite films under different laser power density from 10 nJ cm^{-2} to 20 uJ cm^{-2} (Figure 2g and Supplementary Figure 6), are included to demonstrate the variations of the effective PL lifetime and tail lifetimes (Supplementary Figure 7) under different excitation densities.

We also measured the excitation density dependence of the normalized PLQE (Figure 2h). We found that for the samples with Ge molar fractions of 0-20%, the generally flat PL lifetime and PLQE curves at excitation fluences below 100 nJ cm^{-2} indicate an excitonic character of the emissive species. The PLQEs reduce at higher fluences, in line with possible Auger-like processes under these conditions.

Figure 2i and Supplementary Figure 8 are included to demonstrate the stability of the perovskite films under different excitation densities, and the result shows that the Ge substitution could effectively improve the film stability.

Supplementary Figure 11 and Supplementary Figure 12 show the J-V characteristics of the hole- and electron-only devices based on Ge-Pb perovskite films. Compared to the control devices, the efficiency improvement of the Ge-Pb PeLEDs with 10 mol% Ge can be attributed to lower trap densities in these films from space-charge limited current (SCLC) analysis (Supplementary Figure 11a-b and Supplementary Figure 12a-b).

Please see highlighted sections in the revised text for further discussion on these new measurements.

Reviewer #3 (Remarks to the Author):

In this article, Dawei Di and coworkers used Ge-Pb based green perovskites to produce eco-friendly perovskite LEDs. While it is important to reduce the Pb-content to address the toxicity, I believe the data shown here do not provide a concrete evidence on reduced Pb in their perovskite structures. In addition, the EQEs are substantially lower compared to many other literature articles. Therefore, I would recommend a major revision before the publication in Nature Communications.

R: We appreciate the professional and constructive comments from the reviewer. Taking these comments on board, we have carried out major revisions to improve the paper. We hope the reviewer would agree that the revised paper is acceptable for publication.

My comments are given below

1. There is no experimental evidence on the presence of 30% Ge in their perovskites. From XRD we overestimate this data. Therefore, I would recommend authors to perform rigorous elemental analyses (ICP-MS/OES or AAS) before claiming 30% Ge.

R: We thank the reviewer for raising this important point. We agree that the presence of Ge in the perovskite films should be supported by further experimental evidence. To address this point, we performed inductively coupled plasma optical emission spectroscopy (ICP-OES) measurements to determine the actual Ge fraction in the films. The results are shown in Figure 1a in the revised manuscript. In the revised text, we clarify that:

The actual molar fractions of Ge in the resultant thin films are generally in line with that in the precursor solution, as confirmed by inductively coupled plasma optical emission spectroscopy (ICP-OES) (Figure 1a). The molar fractions of Ge in the films are slightly higher than the intended values when the molar fractions of Ge precursor are equal to or less than 20% (For 10% and 20% Ge content in precursor, the corresponding Ge fractions in films are 15.0% and 25.7%, respectively). When the Ge fractions in precursor exceed 30%, the Ge fractions in films show very close agreement.

2. I am highly doubtful that with presence of 30% Ge, the perovskite can still hold its crystal structure. I would derive the tolerance factor to make sure the perovskite is still holding its orthorhombic structure and supporting XRD.

R: Based on the tolerance factor model ($t = \frac{R_A + R_X}{\sqrt{2}(R_B + R_X)}$) and ICP-OES results, we found that increasing the fraction of Ge substitution in the perovskite structure tends to improve the tolerance factor (t) of the Ge-Pb perovskite films, as shown in the Table below. This implies that the Ge-Pb perovskite are able to hold its orthorhombic structure.

Table| The tolerance factor (t) of perovskite films with different Ge content.

Ge molar fraction in solutions	Ge molar fraction in solid films (ICP-OES results)	Tolerance factor (t)
0	0	0.81
10%	14.98%	0.83
20%	25.68%	0.85
30%	30.12%	0.85
40%	40.68%	0.87
50%	50.32%	0.88

In consideration of the reviewer's comments, we have carried out Rietveld refinement of the XRD data in greater precision. The updated results are shown in Figure 1c and Supplementary Figure 1. From the new Rietveld refinement results, we found that some Ge ions exist in the 2D perovskite structure while the rest are in the 3D perovskite lattice. While the ICP-OES and XRD results both confirm the Ge ions are present in the quasi-2D/3D perovskite nanocrystals, at this stage we are unable to quantify the exact fractions of Ge in the 2D and 3D perovskite phases separately due

to the limited precision of the XRD measurements.

Measurements with further improved precision will be carried out in our future studies to identify the exact positions of the Ge ions in the quasi-2D/3D perovskite structure. In the revised paper, we have clarified that from the Rietveld refinement results, Ge are expected to exist in the quasi-2D/3D perovskite structures (Page 3, highlighted).

3. From Figure 2e, why slower decay in 40% Ge samples, although the PLQY is low, compared to samples with 20% is unclear. I would recommend the authors to present this at various incident power densities and see if the trends change. I appreciate that the authors admit this discrepancy, however, providing more insights on decay kinetics will certainly add a value to this paper.

R: The reviewer has raised a very good point. Following the reviewer's suggestion, we have carried out power-dependent (10 nJ cm^{-2} to 20 uJ cm^{-2}) transient PL decay measurements in this revision. The results are shown in Figure 2g and Supplementary Figure 6 of the revised paper. The effective PL lifetime and tail lifetimes under different excitation densities are shown in Supplementary Figure 7 in the revised manuscript. Transient PL measurements presented in the previous version of the paper has also been re-done in a more careful manner to ensure the reliability of our observation.

Further discussions on these new measurements are presented in the revised paper (see page 5-6 highlighted).

4. Why EQEs of their control devices are lower compared to equivalent LEDs in the literature?

R: To address this point, we have carried out further optimization for the Ge-free control devices, a peak EQE of $\sim 11.3\%$ has been obtained. This result agrees well with the EQEs from similar device structures and emissive layer compositions as reported in the literature (see e.g. Chemistry of Materials 31:83 (2019), which shows EQEs of 10.1% for a similar device structure of ITO/PEDOT:PSS/PVK/perovskite/TPBi/Al). We have also achieved higher efficiencies for the champion Ge-Pb PeLEDs as a result of improved device optimization during the revision process.

5. Stability is an important factor. So, I would recommend authors to present operational stability with time, and curious to see how it changes when Ge is present.

R: We thank the reviewer for the suggestion. The stability of PL and EL with different Ge molar fractions has been tested.

In contrast to the improved PL stability for films under optical excitation (Figure 2i and Supplementary 8), the stability of our Ge-Pb PeLEDs with 10 and 20 mol% Ge

content (T_{50} for EL: 10 and 18 min at 1 mA/cm²) is inferior to that of the Ge-free control device (T_{50} for EL: ~30 min) (Supplementary Figure 13a). Improving the device stability is an important subject of our future research.

REVIEWERS' COMMENTS

Reviewer #1 (Remarks to the Author):

The authors have addressed the main issues that had been raised, chiefly the determination of the Ge fraction in their materials. The additional data and analysis convincingly demonstrate the main claim. I stand by the overall assessment of an interesting topic and a solid piece of experimental work.

I therefore recommend publication.

Reviewer #2 (Remarks to the Author):

Many new data are included in the revised manuscript. Most of the comments are replied in detail, and the provided data and explanations are convincing. I think that this work is ready for publication.

Reviewer #3 (Remarks to the Author):

I am fairly convinced that Ge is entering in the perovskite lattice, and satisfied with author's response to most of my comments. Therefore, I would recommend this work publication in Nature Comm.

Response to Review Comments

Reviewer #1 (Remarks to the Author):

The authors have addressed the main issues that had been raised, chiefly the determination of the Ge fraction in their materials. The additional data and analysis convincingly demonstrate the main claim. I stand by the overall assessment of an interesting topic and a solid piece of experimental work.

I therefore recommend publication.

Reviewer #2 (Remarks to the Author):

Many new data are included in the revised manuscript. Most of the comments are replied in detail, and the provided data and explanations are convincing. I think that this work is ready for publication.

Reviewer #3 (Remarks to the Author):

I am fairly convinced that Ge is entering in the perovskite lattice, and satisfied with author's response to most of my comments. Therefore, I would recommend this work publication in Nature Comm.

R: We thank the reviewers for their valuable comments, and for recommending publication of our paper in *Nature Communications*.